# Txnip deletions and missense alleles prolong the survival of cones in a retinitis pigmentosa mouse model

Yunlu Xue[1,2], Yimin Zhou[2,3], Constance L Cepko[1,4]*

[1]Departments of Genetics and Ophthalmology, Blavatnik Institute, Harvard Medical School, Boston, United States; [2]Lingang Laboratory, Shanghai, China; [3]School of Life Science and Technology, ShanghaiTech University, Shanghai, China; [4]Howard Hughes Medical Institute, Boston, United States

*For correspondence: cepko@genetics.med.harvard. edu

Competing interest: The authors declare that no competing interests exist.

**Abstract** Retinitis pigmentosa (RP) is an inherited retinal disease in which there is a loss of cone-mediated daylight vision. As there are >100 disease genes, our goal is to preserve cone vision in a disease gene-agnostic manner. Previously we showed that overexpressing TXNIP, an α-arrestin protein, prolonged cone vision in RP mouse models, using an AAV to express it only in cones. Here, we expressed different alleles of *Txnip* in the retinal pigmented epithelium (RPE), a support layer for cones. Our goal was to learn more of TXNIP's structure-function relationships for cone survival, as well as determine the optimal cell type expression pattern for cone survival. The C-terminal half of TXNIP was found to be sufficient to remove GLUT1 from the cell surface, and improved RP cone survival, when expressed in the RPE, but not in cones. Knock-down of HSP90AB1, a TXNIP-interactor which regulates metabolism, improved the survival of cones alone and was additive for cone survival when combined with TXNIP. From these and other results, it is likely that TXNIP inter-acts with several proteins in the RPE to indirectly support cone survival, with some of these interac-tions different from those that lead to cone survival when expressed only in cones.

## eLife assessment

This **fundamental** study advances our understanding of the cell specific treatment of cone photoreceptor degeneration by Txnip. The evidence supporting the conclusions is **compelling** with rigorous genetic manipulation of Txnip mutations. The work will be of broad interest to vision researchers, cell biologists and biochemists.

## Introduction

Retinitis pigmentosa (RP) is an inherited retinal degenerative disease that affects one in ~4000 people worldwide (*Hartong et al., 2006*). The disease first manifests as poor night vision, likely due to the fact that many RP disease genes are expressed in rod photoreceptors, which initiate night vision. Cone photoreceptors, which are required for daylight, color, and high acuity vision, also are affected, as are the retina-pigmented epithelial (RPE) cells (*Chrenek et al., 2012*; *Napoli et al., 2021*; *Napoli and Strettoi, 2023*; *Wu et al., 2021*), which support both rod and cone photoreceptors. However, cones and RPE cells typically do not express RP disease genes. Nonetheless, RP cones lose function and die after most of the rods in their immediate neighborhood die. While it is not entirely clear what causes cone death, there are data suggesting problems with metabolism, oxidative stress, lack of trophic factors, oversupply of chromophore, and inflammation (*Komeima et al., 2006*; *Mohand-Said et al., 1998*; *Punzo et al., 2009*; *Xue et al., 2023*; *Zhao et al., 2015*). We have been pursuing gene

therapy to address some of these problems. Our hope is to create therapies that are disease-gene agnostic by targeting common problems for cones across disease gene families. One of our strategies is aimed at cone metabolism. Several lines of evidence suggest that RP cones do not have enough glucose, their main fuel source (Reviewed in *Xue and Cepko, 2023*). We found that overexpression of TXNIP, an α-arrestin protein with multiple functions, including glucose metabolism, prolonged the survival of cones and cone-mediated vision in three RP mouse strains (*Xue et al., 2021*). Regarding the mechanism of rescue, we found that it relied upon the utilization of lactate by cones. In addition, cones treated with Txnip showed improved mitochondrial morphology and function. As TXNIP is known to bind directly to thioredoxin, we tested a *Txnip* allele with a single amino acid (aa) change, C247S, which abolishes the interaction with thioredoxin (*Patwari et al., 2006*). This allele provided better rescue than the wild-type (wt) *Txnip* allele, ruling out its interaction with thioredoxin as required for cone rescue. These findings inspired us to further modify Txnip in various ways to look for better rescue, as well as to explore potential mechanisms for Txnip's action. To this end, we also tested a related α-arrestin protein, as well as an interacting partner, for rescue effects.

## Results

### *Arrdc4* reduces *rd1* cone survival

As TXNIP is a member of the α-arrestin protein family, we explored whether another family member might prolong RP cone survival. There are six known α-arrestins in mammals (*Puca and Brou, 2014*). Among them, arrestin domain-containing protein 4 (ARRDC4) is the closest to TXNIP in amino acid sequence, sharing ~60% similar amino acids with TXNIP (*Figure 1A*). ARRDC4 is thought to have functions that are similar to those of TXNIP in regulating glucose metabolism in vitro (*Patwari et al., 2009*). Like TXNIP and other α-arrestins, ARRDC4 is composed of three domains: a N-terminal arrestin (Arrestin N-) domain, a C-terminal arrestin (Arrestin C-) domain, and an intrinsically disordered region (IDR) at the C-terminus. Because an IDR lacks a stable 3D structure under physiological conditions, previous studies using crystallography did not reveal the full structure of the TXNIP protein (*Hwang et al., 2014*). None of the other α-arrestins have been characterized structurally. To begin to examine potential similarities in structure among some of these family members, we utilized an artificial intelligence (AI) algorithm, AlphaFold-2, to visualize the predicted 3D full structure of ARRDC4 (*Jumper et al., 2021*). Similar to TXNIP, ARRDC4 is predicted to have a 'W' shaped arrestin structure, which is composed of the Arrestin N- and C-domains, plus a long IDR which looks like a tail (*Figure 1B*).

 *Arrdc4* was tested for its ability to prolong cone survival in *rd1* mice using AAV-mediated gene delivery, as was done for *Txnip* previously (*Xue et al., 2021*). Expression of *Arrdc4* was driven by a cone-specific promoter, RO1.7, derived from human red opsin (*Krol et al., 2010*; *Wang et al., 1992*; *Ye et al., 2016*). The vector was packaged into the AAV8 serotype capsid. AAV-Arrdc4 was injected sub-retinally into P0 *rd1* mouse eyes along with AAV-H2BGFP, which is used to trace the infection and to label the cone nuclei for counting. At P50, the treated retinas were harvested and flat-mounted for further quantification of cones within the central retina, the area that first degenerates. Unlike Txnip, the cone counts were much lower in Arrdc4 treated retina relative to the AAV-H2BGFP control (*Figure 1C and D*).

### Evaluation of cone survival using *Txnip* deletion alleles expressed in the RPE

We previously showed (*Xue et al., 2021*) that overexpressing the *Txnip* wt allele in the RPE using an RPE-specific promoter, derived from the human *BEST1* gene (*Esumi et al., 2009*), did not improve RP cone survival. The wt allele removes the glucose transporter from the plasma membrane, thus preventing the RPE from taking up glucose for its own metabolism, and preventing it from serving as a conduit for glucose to flow from the blood to the cones. However, a triple mutant, Txnip.C247S.LL351 and 352AA, improved cone survival when expressed only in the RPE (*Xue et al., 2021*). The C247S mutation eliminates the interaction with thioredoxin, and enhances the Txnip rescue when expressed in cones (*Xue et al., 2021*). The LL351 and 352AA mutations eliminate a clathrin-binding site, which is required for Txnip's interaction with clathrin-coated pits for removal of GLUT1 from the cell surface (*Wu et al., 2013*). We previously proposed a model in which Txnip.C247S.LL351 and 352AA promotes the use of lactate by the RPE (*Xue et al., 2021*), as we found was the case when Txnip was expressed

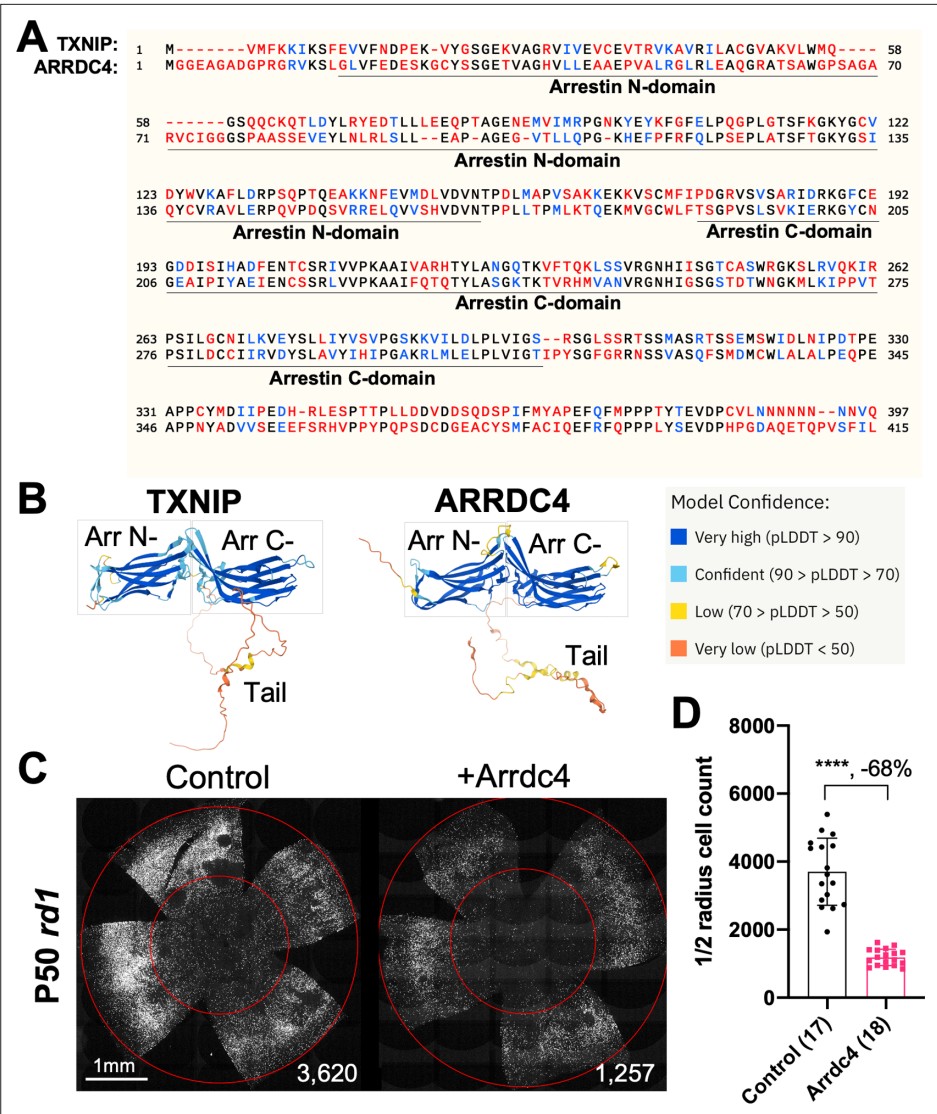

**Figure 1.** Effect of arrestin domain containing protein 4 (Arrdc4) on cone survival in retinitis pigmentosa mice. (**A**) Amino acid sequences of mouse TXNIP and mouse ARRDC4. In the full-length alignment (421 amino acid), Identity: 172/421, 40.86%; Similarity: 246/421, 58.43%; Gaps: 28/421, 6.65%. Color code: identical, black; similar, blue; not similar, red. (**B**) Predicted 3D protein structures of mouse TXNIP and mouse ARRDC4 by artificial intelligence (AI) algorithm AlphaFold-2. Abbreviations: Arr N-, N-terminal arrestin domain; Arr C-, C-terminal arrestin domain. (**C**) Representative P50 *rd1* flat-mounted retinas after P0 subretinal infection with AAV8-RO1.7-Arrdc4 (1×10⁹ vg/eye), plus AAV8-RedO-H2BGFP (2.5×10⁸ vg/eye), or control eyes infected with AAV8-RedO-H2BGFP, 2.5×10⁸ vg/eye alone. (**D**) Quantification of H2BGFP-positive cones within the center of P50 *rd1* retinas transduced with Arrdc4, and control (same as in C). The number in the round brackets '()' indicates the number of retinas within each group. Error bar: standard deviation. Statistics: two-tailed unpaired Student's t-test. **** p<or << 0.0001. RedO: red opsin promoter; RO1.7: a 1.7 kb version of red opsin promoter. AAV: adeno-associated virus.

The online version of this article includes the following source data for figure 1:

**Source data 1.** This file contains the source data of *Figure 1D*.

in cones. Although the RPE normally uses lactate in wt animals, in RP, it is hypothesized that it retains the glucose that it normally would deliver to cones (Reviewed in *Hurley, 2021*). The retention of glucose by the RPE is thought to be due to a reduction in lactate supply, as rods normally provide lactate for the RPE, and with rod loss that source would be greatly diminished. If the RPE can utilize lactate in RP, perhaps using lactate supplied by the blood, and the LL351 and 352AA mutation impairs the ability of TXNIP to remove the glucose transporter from the plasma membrane, this allele of *Txnip*

may then allow glucose to flow from the blood to the cones via the GLUT1 transporter. The expression of Txnip.C247S.LL351 and 352AA allele thus has the potential to address the proposed glucose shortage of RP cones. However, we noted two caveats. One is that the survival of cones was not as robust as when *Txnip* was expressed directly in cones. In addition, the *rd1* retina in the FVB strain used here, even without any treatment, shows holes in the cone layer, which appear as 'craters.' An RP rat model presents a similar pattern (*Ji et al., 2014*; *Ji et al., 2012*; *Zhu et al., 2013*). When Txnip.C247S. LL351 and 352AA are expressed in the RPE, there are more craters in the photoreceptor layer. We note that these craters are common only in the *rd1* allele on the FVB background, i.e., not as common on other inbred mouse strains that also harbor the *rd1* allele, so the meaning of this observation is unclear.

Arrestins are well-known for their protein-protein interactions via different domains. Different regions of TXNIP are known to directly associate with different protein partners to affect several different functions. For example, the N-terminus is sufficient to interact with KPNA2 for TXNIP's localization to the nucleus (*Nishinaka et al., 2004*), while the C-terminus of TXNIP is critical for interactions with COPS5, to inhibit cancer cell proliferation (*Jeon et al., 2005*). The C-terminus of TXNIP is also necessary for inhibition of glycolysis, at least in vitro, through an unclear mechanism (*Patwari et al., 2009*). Based on these studies, we made several deletion alleles of *Txnip*, and expressed them in the RPE using the Best1 promoter. We assayed their ability to clear GLUT1 from the RPE surface (*Figure 2A*), as well as promote cone survival (*Figure 2B–G*). To enable automated cone counting and trace the infection, we co-injected an AAV (AAV8-RedO-H2BGFP-WPRE-bGHpA) encoding an allele of GFP fused to histone 2B (H2BGFP), which localized to the nucleus. As the red opsin promoter was used to express this gene, H2BGFP was seen in cone nuclei, but not in the RPE, if AAV8-RedO-H2BGFP-WPRE-bGHpA was injected alone. However, when an AAV that expressed in the RPE, i.e., AAV8-Best1-Sv40intron-(Gene)-WPRE-bGHpA, was co-injected with AAV8-RedO-H2BGFP-WPRE-bGHpA, H2BGFP was expressed in the RPE, along with expression in cones (*Figure 2A*). We speculate that this is due to concatenation or recombination of the two genomes, such that the H2BGFP comes under the control of the RPE promoter. This may be due to the high copy number of AAV in the RPE, as it did not happen in the reverse combination, i.e., AAV with an RPE promoter driving GFP and a cone promoter driving another gene. It was previously observed that the AAV genome copy number was »10 fold lower in cones than in the RPE (*Wang et al., 2020*).

To assay GLUT1, we focused on the basal surface of the RPE, as it is easier to score than the apical surface, where its processes are intertwined with those of the retina, where GLUT1 is also expressed. The 149-397aa portion of Txnip.C247S (C.Txnip.C247S) had the highest activity for GLUT1 removal from the RPE basal surface in vivo, while the 1-228aa portion (N.Txnip) failed to remove GLUT1 (*Figure 2A* and *Figure 2—figure supplement 1*). As predicted by ColabFold, an AI algorithm based on AlphaFold-2 (*Mirdita et al., 2022*), the Arrestin C-domain, which is part of C.Txnip.C247S, but is not present in the N-domain of TXNIP, interacts with the intracellular C-terminal IDR of GLUT1 (*Figure 2—figure supplement 2*). These results are consistent with these predictions, in that the C-terminal portion of TXNIP is sufficient to bind and clear GLUT1 from cell surface, while the N-domain is not.

Cone survival was assayed in vivo following infection of *rd1* with these missense and deletion alleles at P0 and sacrifice at P50 (*Figure 2B–G*). Similar to Best1-wt Txnip (*Xue et al., 2021*), Best1-Txnip. C247S did not show significant improvement of cone survival, ruling out the C247S mutation alone as promoting the cone survival by Best1-Txnip.C247S.LL351 and 352AA. In addition, Best1-N.Txnip (1-228aa) and Best1-sC.Txnip (255-397aa, sC: short C-) failed to improve cone survival. However, Best1-C.Txnip.C247S (149-397aa), Best1-C.Txnip.C247S.LL351 and 352AA (149-397aa), and Best1-nt. Txnip.C247S[320] (1-320aa, nt: no-tail) promoted significant cone survival compared to the corresponding control retinas. Best1-N.Txnip and Best1-sC.Txnip-treated *rd1* retina did not have increased numbers of craters, while all other vectors increased the number of craters. These results suggest that the C-terminal portion of TXNIP expressed in the RPE is required for RP cone survival, for a function(s) that is unrelated to the removal of GLUT1, or to the mechanism that leads to an increase in craters.

## Evaluation of *Txnip* deletion alleles for autonomous cone survival

Our previous study used the human red opsin promoter, 'RedO,' in AAV to drive the expression of Txnip in *rd1* cones, with a low level of expression in some rods. This same strategy was used to

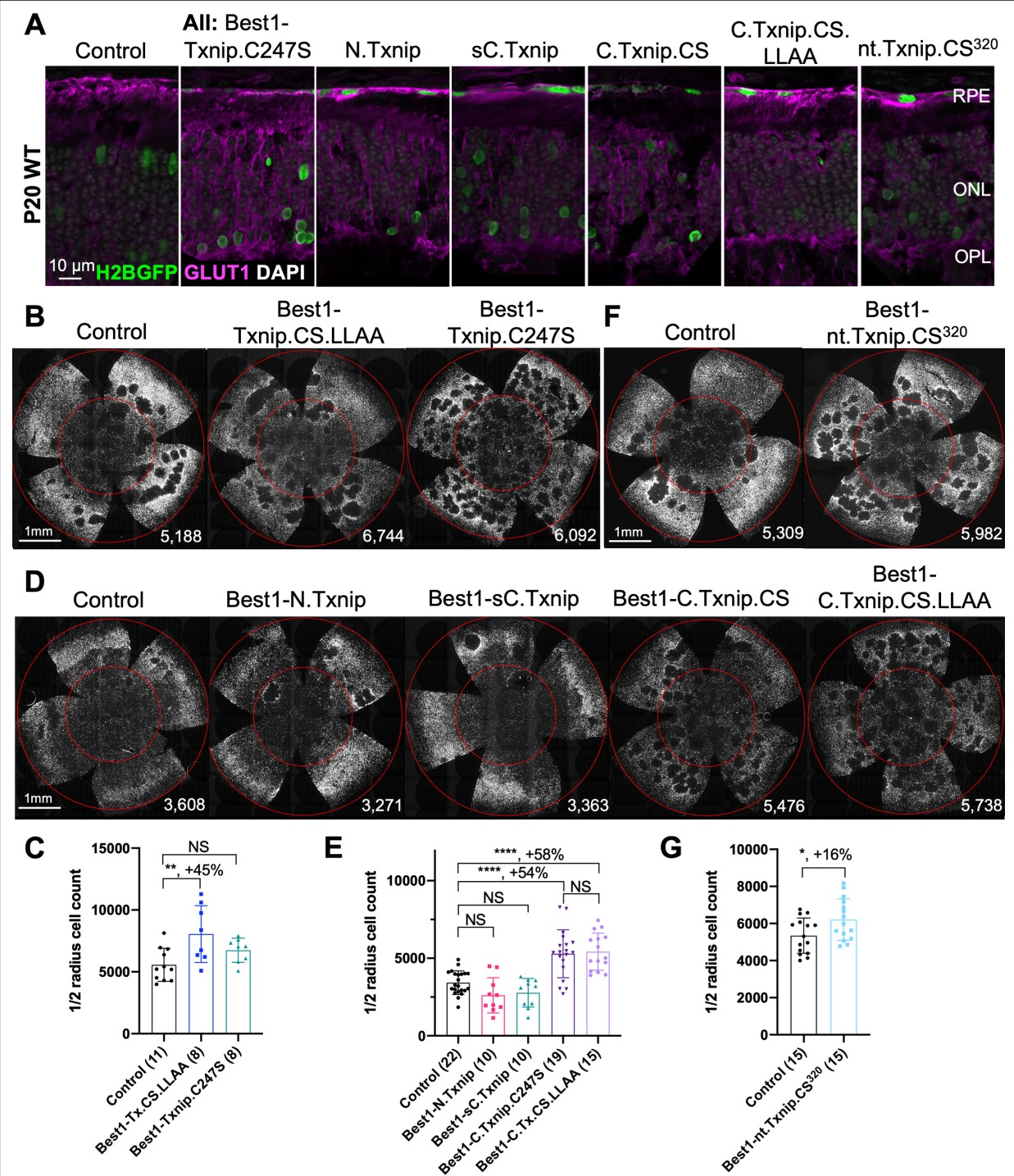

**Figure 2.** *Txnip* deletions expressed only within retinal pigmented epithelium (RPE) cells: effects on GLUT1 removal and cone survival. (**A**) Glucose transporter 1 (GLUT1) expression in P20 wild-type eyes infected with control (AAV8-RedO-H2BGFP, $2.5×10^8$ vg/eye), or a *Txnip* allele ($2.5×10^8$ vg/eye) plus RedO-H2BGFP ($2.5×10^8$ vg/eye), as indicated in each panel. *Txnip* deletions are detailed in **Figure 4**. GLUT1 intensity from basal RPE is quantified in **Figure 2—figure supplement 1**. Magenta: GLUT1; green: RedO-H2BGFP for infection tracing; gray: DAPI. (**B, D, F**) Representative P50 *rd1* flat-mounted retinas after P0 infection with one of seven different *Txnip* alleles expressed only within the RPE, as indicated in the figure, or control eyes infected with AAV8-RedO-H2BGFP, $2.5×10^8$ vg/eye alone. (**C, E, G**) Quantification of H2BGFP-positive cones within the center of P50 *rd1* retinas transduced with indicated vectors, as shown in B, D, F. The number in the round brackets '()' indicates the number of retinas within each group. Error bar: standard deviation. Statistics: ANOVA and Dunnett's multiple comparison test for C and E; two-tailed unpaired Student's t-test for G. C.Txnip.CS:

*Figure 2 continued on next page*

*Figure 2 continued*

C-terminal portion of Txnip.C247S; C.Txnip.CS.LLAA: C-terminal portion of Txnip.C247S.LL351 and 352AA; nt.Txnip.CS³²⁰: no tail Txnip (1-320aa). NS: not significant, p>0.05, *p<0.05, **p<0.01, ****p<or << 0.0001. Best1: Best1 promoter.

The online version of this article includes the following source data and figure supplement(s) for figure 2:

**Source data 1.** This file contains the source data of *Figure 2C, E and G* and *Figure 2—figure supplement 1*.

**Figure supplement 1.** *Txnip* deletions expressed only within retinal pigmented epithelium (RPE) cells: quantification of the Glucose transporter 1 (GLUT1) level within the basal surface of the RPE.

**Figure supplement 2.** Predicted protein-protein interactions of TXNIP and Glucose transporter 1 (GLUT1) by an algorithm, ColabFold, based on AlphaFold-2.

evaluate whether the aforementioned deletion alleles of *Txnip* could prolong cone survival. Neither N.Txnip (1-228aa) nor C.Txnip.C247S (149-397aa) promoted significant improvement in *rd1* cone survival. However, nt.Tnxip.C247S³⁰¹ (1-301aa) and nt.Txnip.C247S³²⁰ (1-320aa) promoted survival of *rd1* cones: 47% and 63% more cones than the control GFP virus, respectively (*Figure 3A and B*).

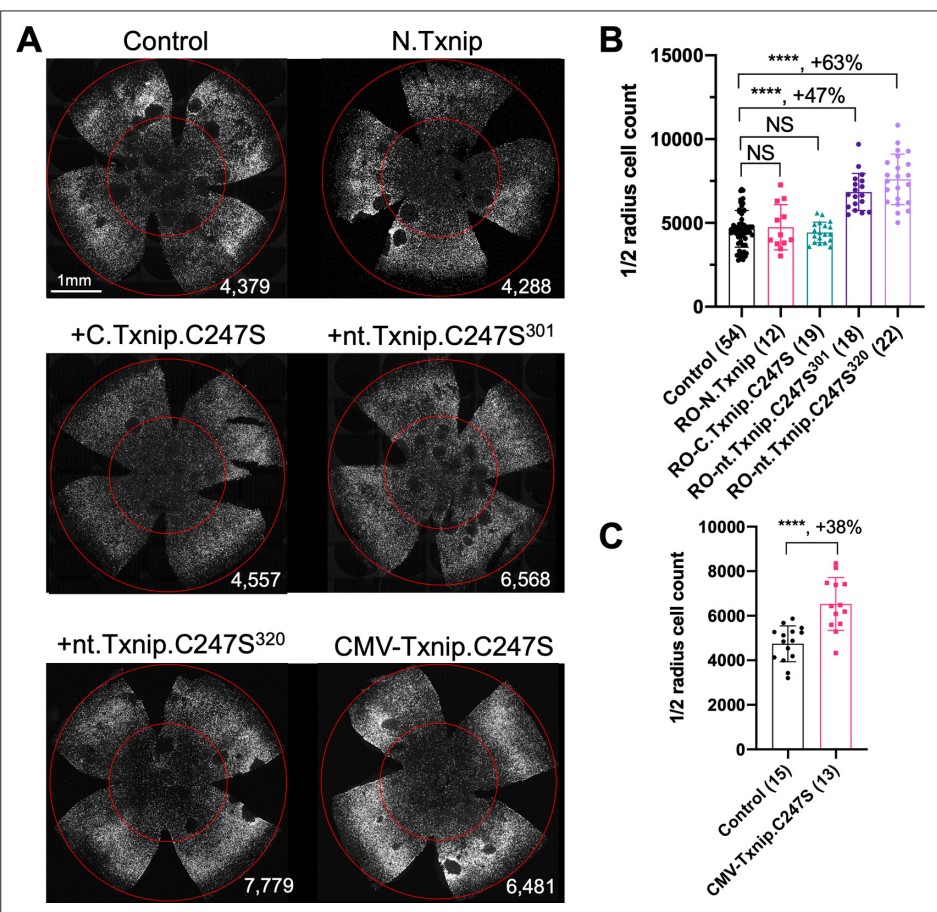

**Figure 3.** Tests of *Txnip* alleles on cone survival. (**A**) Representative P50 *rd1* flat-mounted retinas after P0 infection with 1 of 5 different *Txnip* alleles (AAV8-RedO- N.Txnip /C.Txnip.C247S/ nt.Txnip.C247S¹⁻³⁰¹/nt.Txnip.C247S¹⁻³²⁰ or AAV8-CMV-Txnip.C247S, ≈1×10⁹ vg/eye, plus AAV8-RedO-H2BGFP, 2.5×10⁸ vg/eye), or control eyes infected with AAV8-RedO-H2BGFP, 2.5×10⁸ vg/eye alone. (**B, C**) Quantification of H2BGFP-positive cones within the center of P50 *rd1* retinas transduced with AAV8-RedO- N.Txnip /C.Txnip.C247S/ nt.Txnip.C247S¹⁻³⁰¹/nt.Txnip.C247S¹⁻³²⁰ or AAV8-CMV-Txnip.C247S, and control (same as in A). The number in the round brackets '()' indicates the number of retinas within each group. Error bar: standard deviation. Statistics: ANOVA and Dunnett's multiple comparison test for B; two-tailed unpaired Student's t-test for C. NS: not significant, ****p<or << 0.0001.

The online version of this article includes the following source data for figure 3:

**Source data 1.** This file contains the source data of *Figure 3B and C*.

In comparison, the full-length Txnip.C247S promoted an increase of 97% in cones in our previous study (*Xue et al., 2021*). These results show that the full-length Txnip provides the most benefit in terms of RP cone survival. To determine if expression of this allele might give increased survival when expressed in both the RPE and in cones, we used a CMV promoter to drive expression, as CMV

| | RPE GLUT1 Removal | Cone Rescue: Expression in RPE | Cone Rescue: Expression in Cones |
|---|---|---|---|
| wt Txnip (1-397aa) | Y (-66%) | N | Y (60%) |
| Txnip.S308A (1-397aa) | NT | NT | N |
| Txnip.C247S (1-397aa) | Y (-34%) | N | Y (97%) |
| Txnip.C247S.LL351&352AA (1-397aa) | N | Y (45%) | Y (43%) |
| N.Txnip (1-228aa) | N | N | N |
| C.Txnip.C247S (149-397aa) | Y (-74%) | Y (54%) | N |
| C.Txnip.C247S.LL351&352AA (149-397aa) | N | Y (58%) | NT |
| sC.Txnip (255-397aa) | Y (-28%) | N | NT |
| nt.Txnip.C247S (1-301aa) | NT | NT | Y (47%) |
| nt.Txnip.C247S (1-320aa) | N | Y (16%) | Y (63%) |
| | Cone Rescue | | |
| CMV-Txnip.C247S | Y (38%) | | |
| RedO-Txnip.C247S + Best1-Nrf2 | Y (14% increase to RedO-Txnip.C247S) | | |

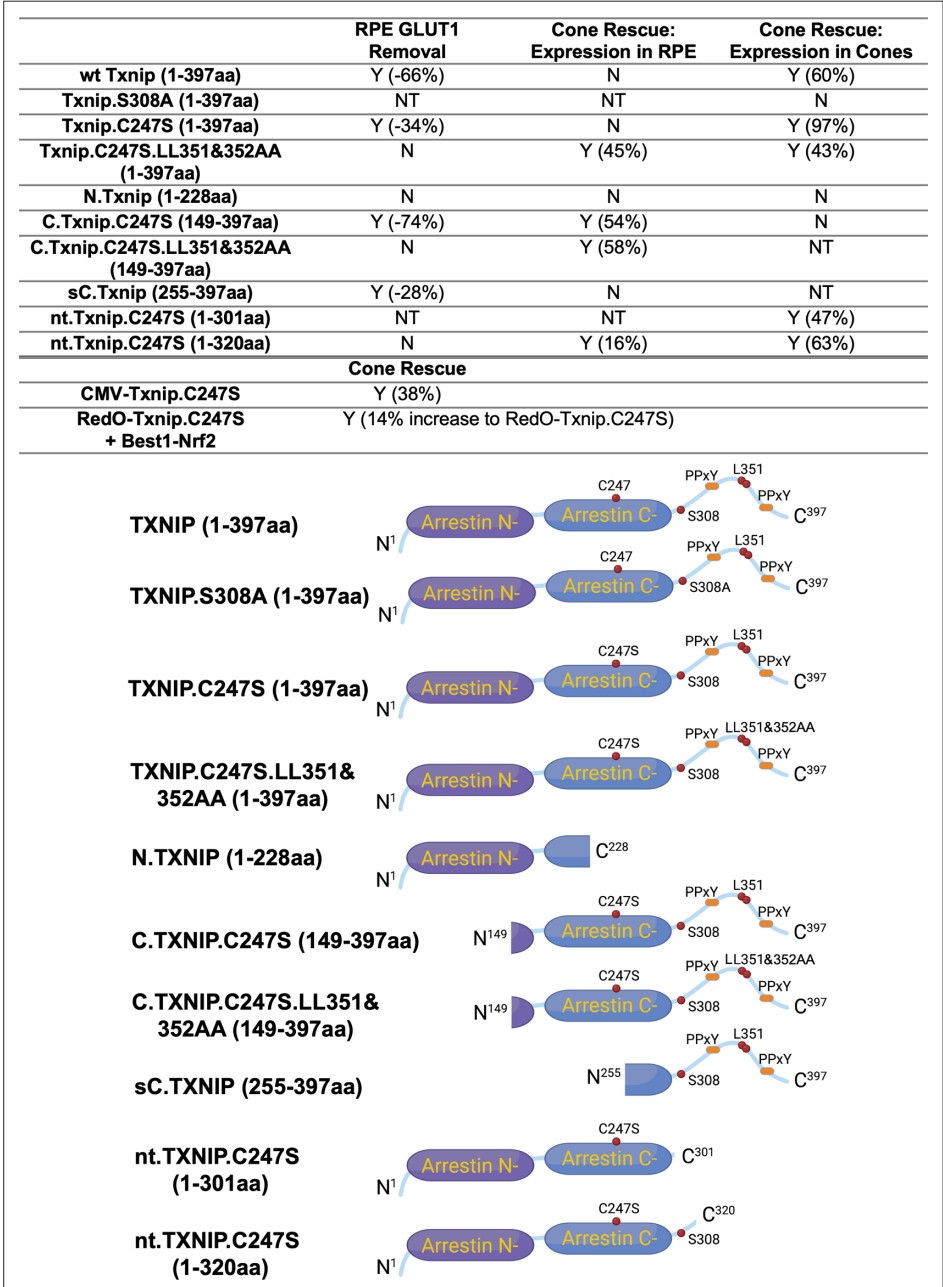

**Figure 4.** Summary of various alleles of *Txnip* in this and previous study (*Xue et al., 2021*). 'Retinal pigmented epithelium (RPE) Glucose transporter 1 (GLUT1) Removal' refers to the amount of GLUT1 immunohistochemical signal on the basal surface following expression in the RPE using the Best1 promoter. 'Cone Rescue: Expression in RPE' refers to cone rescue following expression only in the RPE using the Best1 promoter. 'Cone Rescue: Expression in Cones' is due to expression only in cone photoreceptors using the RedO promoter. Abbreviations: Y (x%): Yes with x% increase compared to AAV-H2BGFP control; N: No; NT: Not tested. N.TXNIP, N-terminal portion of TXNIP; C.TXNIP.C247S, C-terminal portion of TXNIP.C247S mutant allele; sC.TXNIP: a shorter version of C-terminal portion of TXNIP; nt.TXNIP.C247S, no tail version TXNIP.C247S mutant allele; Arrestin N-, N-terminal arrestin domain; Arrestin C-, C-terminal arrestin domain; PPxY, a motif where P is proline, x is any amino acid and Y is tyrosine.

expresses highly in both cell types (*Xiong et al., 2015*). CMV-Txnip.C247S provided a 38% rescue (*Figure 3A and C*), which is lower than RedO-Txnip.C247S (97%) alone. These and previous results are summarized in *Figure 4*.

## Inhibiting *Hsp90ab1* prolongs *rd1* cone survival

To further investigate the potential mechanism(s) of cone survival induced by Txnip, we considered the list of protein interactors that were identified in HEK293 cells using biotinylated protein interaction pull-down assay plus mass spectrometry (*Forred et al., 2016*). Forred et al. identified a subset of proteins that interact with Txnip.C247S, the mutant that provides better cone rescue than the wt *Txnip* allele (*Xue et al., 2021*). As we found that Txnip promotes the use of lactate in cones, and improves mitochondrial morphology and function, we looked for TXNIP interactors that are relevant to mitochondria. We identified two candidates, PARP1 and HSP90AB1. PARP1 mutants have been shown to protect mitochondria under stress (*Hocsak et al., 2017*; *Szczesny et al., 2014*). Accordingly, in our previous study, we crossed the null PARP1 mice with *rd1* mice, to ask if mitochondrial improvements alone were sufficient to induce cone rescue. We found that it was not. In our current study, we thus prioritized HSP90AB1 inhibition, which had been shown to improve skeletal muscle mitochondrial metabolism in a diabetes mouse model (*Jing et al., 2018*).

Three shRNAs targeting different regions of the mRNA of *Hsp90ab1* (shHsp90ab1) were delivered by AAV into the retinas of wt mice. Knock-down was evaluated using an AAV encoding a FLAG-tagged HSP90AB1 that was co-injected with the AAV-shRNA. All three shRNAs reduced the HSP90AB1-FLAG signal compared to the shNC, the non-targeting control shRNA (*Figure 5A and B*), suggesting that they are able to inhibit the expression of HSP90AB1 protein in vivo. The promotion of cone survival was then tested in *rd1* mice using these shRNA constructs. The two shRNAs with the most activity in reducing the FLAG-tagged HSP90AB1 signal, shHsp90ab1[#a], and shHsp90ab1[#c], were found to increase the survival of *rd1* cones at P50 (*Figure 5C and D*). To determine if this effect was capable of increasing the Txnip rescue, the shRNAs were co-injected with Txnip.C247S. A slight additive effect of shHsp90ab1 and Txnip.C247S was observed (*Figure 5E and F*). We also asked if there might be an effect of the knock-down of *Hsp90ab1* on a *Parp1* loss of function background. We did not observe any rescue effect of the shRNAs on this background (*Figure 5G and H*).

## Discussion

In RP, the RPE cells and cones degenerate due to non-autonomous causes after the death of rods. Although the causes of cone death are not entirely clear, one model proposes that they do not have enough glucose, their main fuel source (*Hurley, 2021*; *Punzo et al., 2009*; *Xue and Cepko, 2023*). In a previous study, we found that Txnip promoted the use of lactate within cones and led to healthier mitochondria. The mechanisms for these effects are unclear, and we sought to determine what domains of TXNIP might contribute to these effects, as well as explore alleles of *Txnip* that might be more potent for cone survival. We further tested the rescue effects of several alleles when expressed in the RPE, a support layer for cones, through which nutrients, such as glucose, flow to the cones from the choriocapillaris. The results suggest that Txnip has different mechanisms for Txnip-mediated cone survival when expressed in the RPE versus in cones.

The C-terminal portion of Txnip.C247S (149-397aa) expressed within the RPE, but not within cones, delayed the degeneration of cones (*Figure 2*). The full-length Txnip.C247S expressed within cones, but not within the RPE, was the most effective configuration for cone survival (*Figure 3*). The expression of full-length Txnip.C247S in both the RPE and cones did not provide better rescue than in cones alone. As TXNIP has several domains that presumably interact with different partners, it is possible that these different effects on cone survival are due to the interaction of different TXNIP domains with different partners in the RPE versus the cones, or different results from the interactions of the same domains and partners in the two cell types. The N-terminal half of TXNIP (1-228aa) might exert harmful effects in the RPE, that negate the beneficial effects from the C-terminal half, suggested by the observation that its removal, in the C-terminal 149–397 allele, led to better cone survival when expressed in the RPE (*Figure 2*). In cones, the C-terminal half, including the C-terminal IDR tail, may cooperate with the N-terminal half, or negate its negative effects, to benefit RP cone survival.

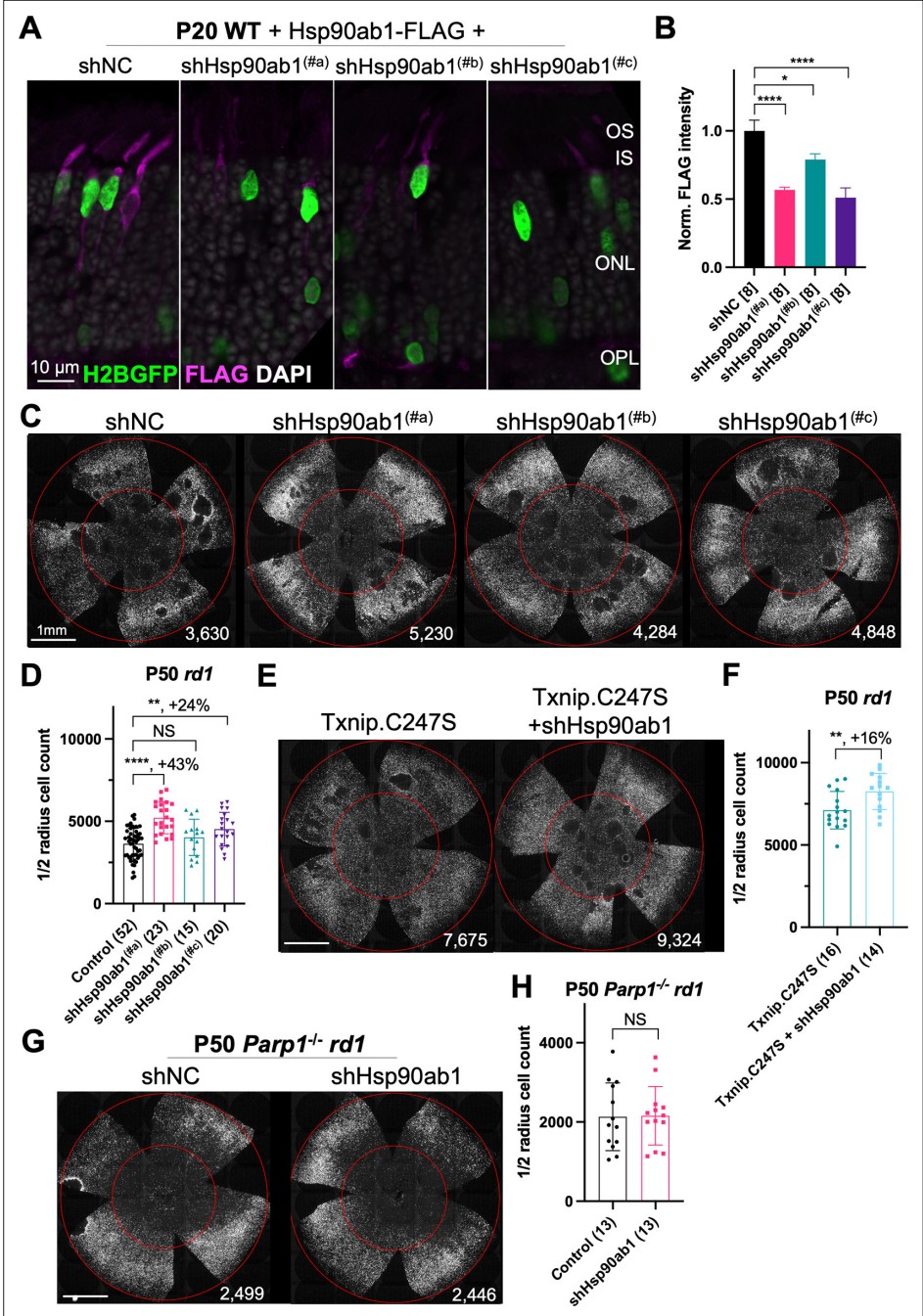

**Figure 5.** Effect of knockdown of *Hsp90ab1* in retinitis pigmentosa cones in vivo. (**A**) AAV8-RO1.7-Hsp90ab1-FLAG (1×10⁹ vg/eye) co-injected with shNC (non-targeting shRNA control, AAV8-RedO-shRNA, 1×10⁹ vg/eye) or co-injected with *Hsp90ab1* shRNAs #a, #b, #c (AAV8-RedO-shRNA, 1×10⁹ vg/eye) in P20 wild-type (wt) retina, all also injected with AAV8-RedO-H2BGFP (2.5×10⁸ vg/eye) to track the infection. Magenta: anti-FLAG; green: anti-GFP; gray: DAPI. Right panel: (**B**) The quantification of FLAG intensity from multiple fields of inner segment regions in A. The number in the square brackets '[]' indicates the number of images taken from regions of interest of one retina, in each condition. (**C**) Representative P50 *rd1* flat-mounted retinas injected with shNC, shHsp90ab1(#a), shHsp90ab1(#b), or shHsp90ab1(#c) (AAV8-RedO-shRNAs, 1×10⁹ vg/eye, plus AAV8-RedO-H2BGFP, 2.5×10⁸ vg/eye). (**D**) Quantification of H2BGFP-positive cones within the center of P50 *rd1* retinas transduced with shNC, shHsp90ab1(#a, #b, #c) (same as in C). (**E**) Representative P50 *rd1* flat-mounted retinas with H2BGFP (gray)-labeled cones transduced with Txnip.C247S or Txnip.C247S+shHsp90ab1 (AAV8-RedO-Txnip.C247S, 1×10⁹ vg/eye; AAV8-RO1.7-shHsp90ab1(#a or #c), 1×10⁹ vg/eye; plus AAV8-RedO-H2BGFP, 2.5×10⁸ vg/eye). (**F**) Quantification of H2BGFP-positive cones within the center of P50 *rd1* retinas transduced with Txnip.

*Figure 5 continued on next page*

*Figure 5 continued*

C247S or Txnip.C247S+shHsp90ab1 (same as in E). (**G**) Representative P50 *Parp1*[-/-] *rd1* flat-mounted retinas with H2BGFP (gray)-labeled cones transduced with shNC (non-targeting shRNA control, AAV8-RedO-shRNA, 1×10⁹ vg/eye; plus AAV8-RedO-H2BGFP, 2.5×10⁸ vg/eye) or shHsp90ab1 (AAV8-RedO-shRNA #a or #c, 1×10⁹ vg/eye; plus AAV8-RedO-H2BGFP, 2.5×10⁸ vg/eye). (**H**) Quantification of H2BGFP-positive cones within the center of P50 *Parp1*[-/-] *rd1* retinas transduced with shNC or shHsp90ab1 (same as in G). Error bar: standard deviation. Statistics: ANOVA and Dunnett's multiple comparison test for B and D; two-tailed unpaired Student's t-test for F and H. NS: not significant, p>0.05, *p<0.05, **p<0.01, ***p<0.001 ****p<or << 0.0001.

The online version of this article includes the following source data and figure supplement(s) for figure 5:

**Source data 1.** This file contains the source data of *Figure 5B, D, F and H*.

**Figure supplement 1.** Predicted 3D protein structures of HSP90AB1 and PARP1.

**Figure supplement 2.** Predicted 3D protein interactions among TXNIP, HSP90AB1, and PARP1 by AI algorithm AlphaFold Multimer from two angles of view.

---

However, the C-terminal half is not sufficient for cone rescue when expressed in cones, as the 149–397 allele did not rescue.

The C-terminal half of TXNIP apparently affects cone survival differently when expressed within the two cell types. This notion is informed by the different rescue effects of expression of the 149–397 allele, which rescues cones when expressed in the RPE, but not when expressed in cones. This domain loses the cone rescue activity if it loses aa 149–254, when expressed in the RPE, as shown by the 255–397 allele. In cones, the rescue activity is present in the 1–301 and the 1–320 allele, but is lost in the 149–397 allele. It is possible that effects on protein structure cause this loss, or that an interaction between N-terminal and C-terminal domains is required for cone rescue within cones.

One TXNIP function that likely is important to these effects in the two cell types is TXNIP's removal of the glucose transporter from the plasma membrane. The LLAA TXNIP mutant is unable to effectively remove the transporter, due to its loss of interaction with clathrin (*Wu et al., 2013*). When this mutant allele is expressed in the RPE, it leads to improved cone survival, in contrast to the wt allele. This might be due to better health in the RPE, when it is able to take up glucose to fuel its own metabolism, and/or to provide glucose to cones. When the LLAA allele is expressed in cones, it also promotes cone survival, though not as well as the wt allele (*Xue et al., 2021*). The wt allele might be more beneficial in cones if it is part of the mechanism that forces cones to rely more heavily on lactate vs. glucose. All of these observations of cone rescue from expression within cones suggest that cone rescue relies on activities that reside in both the N and C-terminal portions, including the ability of TXNIP to interact with clathrin. However, it will be important to probe structural alterations and stability of TXNIP in cones and RPE when these various alleles are expressed to further support these hypotheses.

ARRDC4, the most similar α-arrestin protein to TXNIP that also has Arrestin N- and C- domains, accelerated RP cone death when transduced via AAV (*Figure 1*). This observation suggests that TXNIP has unique functions that protect RP cones. Recently, ARRDC4 has been proposed to be critical for liver glucagon signaling, which could be negated by insulin (*Dagdeviren et al., 2023*). The implication of this potential role regarding RP cone survival is unclear, but interestingly, the activation of the insulin/mTORC1 pathway is beneficial to RP cone survival (*Punzo et al., 2009*; *Venkatesh et al., 2015*).

Regarding potential protein interactions beyond the glucose transporter, the interaction of TXNIP with thioredoxin is apparently negative for cone survival, as we found in our previous study with the C247S allele. This is most easily understood by the release of thioredoxin from TXNIP, whereupon it can play its anti-oxidation role, which would be important in the RP retina which exhibits oxidative damage. It also would free TXNIP to interact with other partners, of which there are several, though many also depend upon C247 (*Forred et al., 2016*). Another partner interaction suggested by previous studies and explored here is the interaction with HSP90AB1 (*Figure 5*). HSP90AB1 interacts with both the wt and C247S alleles (*Forred et al., 2016*). Little is known about the function of HSP90AB1. Knocking down *Hsp90ab1* improved mitochondrial metabolism of skeletal muscle in a diabetic mouse model (*Jing et al., 2018*). Knocking out HSP90AA1, a paralog of HSP90AB1 which has 14% different amino acids, led to rod death and correlated with PDE6 dysregulation (*Munezero et al., 2023*). Inhibiting HSP90AA1 with small molecules transiently delayed cone death in human retinal

organoids under low glucose conditions (*Spirig et al., 2023*). However, the exact role of HSP90AA1 in photoreceptors needs to be clarified, and the implications for HSP90AB1 in RP cones are still unclear.

Here, we found that sh-mediated knock-down of *Hsp90ab1* enhanced cone survival in *rd1* mice. This rescue seems to be dependent on PARP1, another binding partner of wt TXNIP and Txnip. C247S (*Forred et al., 2016*). As shown by PARP1 knock-out mice, PARP1 is deleterious to mitochondrial heath under stressful conditions (*Hocsak et al., 2017*; *Szczesny et al., 2014*; *Xue et al., 2021*). When we examined a possible rescue effect of PARP1 loss on *rd1* cone survival, we did not see a benefit, indicating that the TXNIP-mediated rescue is not due solely to its beneficial effects on mitochondria, nor does TXNIP-mediated rescue rely upon PARP1 (*Xue et al., 2021*). These results indicate that the Txnip rescue is more complex than inhibition of HSP90AB1, and a PARP1-independent mechanism is involved. It is possible that HSP90AB1 directly interacts with PARP1, and this interaction is critical for shHsp90ab1 to benefit RP cones. We looked into the predicted 3D structures of HSP90AB1 and PARP1 using AlphaFold-2 (*Figure 5—figure supplement 1*), but did not gain additional insight into such interactions. We also explored AlphaFold Multimer, which is an algorithm predicting the interaction of multiple proteins based upon AlphaFold-2 (*Evans et al., 2021*), and noticed that the Arrestin-C domain of TXNIP linked PARP1 and HSP90AB1 together in one of the predicted models (*Figure 5—figure supplement 2*). Despite the unclear mechanism, combining *Hsp90ab1* inhibition with Txnip.C247S could be a potential combination therapy to maximize the protection of RP cones.

# Materials and methods

## Key resources table

| Reagent type (species) or resource | Designation | Source or reference | Identifiers | Additional information |
|---|---|---|---|---|
| Antibody | GLUT1 (rabbit monoclonal) | Abcam | ab115730 | IHC (1:500) |
| Genetic reagent (*M. musculus*) | *Arrdc4* cDNA | GeneCopoeia | Cat. #: Mm26972 NCBI: NM_001042592.2 | |
| Genetic reagent (*M. musculus*) | *Hsp90ab1* cDNA | GeneCopoeia | Cat. #: Mm03161 NCBI: NM_008302.3 | |
| Software, algorithm | Protein 3D structure prediction | AlphaFold-2 | TXNIP (*M. musculus*); ARRDC4 (*M. musculus*); HSP90AB1 (*M. musculus*); PARP1 (*M. musculus*) | *Jumper et al., 2021*; https://alphafold.ebi.ac.uk |
| Software, algorithm | Protein 3D interaction prediction | ColabFold | AlphaFold2_mmseqs2 | *Mirdita et al., 2022*; *Ovchinnikov, 2021*; https://github.com/sokrypton/colabfold |
| Software, algorithm | Protein 3D interaction prediction | COSMIC2 | AlphaFold2 – Multimer | *Evans et al., 2021*; http://cosmic-cryoem.org/tools/alphafoldmultimer/ |
| Software, algorithm | Protein 3D structure viewer | RCSB PDB | Mol* 3D Viewer | To visualize the 3D structure of proteins in.pdb files https://www.rcsb.org/3d-view |

The material and methods in this study are similar to those used in our previous study (*Xue et al., 2021*). The cone number of the central retina is defined as the counts of H2BGFP-positive cells within the central portion of the retina. New reagents and algorithms used in this study are listed in the Key resources table above. *Txnip* deletion alleles were cloned from the Txnip plasmid using Gibson assembly (*Figure 4*). The following sense strand sequences were used to knock down the *Hsp90ab1*: shHsp90ab1(#a) 5′- GCATCTACCGCATGATTAAAC-3′; shHsp90ab1(#b) 5′- CCAGAAGTCCATCTAC TATAT-3′; shHsp90ab1(#c) 5′- CCTGAGTACCTCAACTTTATC-3′. In almost all experiments, other than as noted, one eye of the mouse was treated with control (AAV8-RedO-H2BGFP, $2.5×10^8$ vg/eye), and the other eye was treated with the experimental vector plus AAV8-RedO-H2BGFP, $2.5×10^8$ vg/eye. For RPE basal surface GLUT1 quantification, multiple regions of interest (ROI) were selected from at least three eyes of each condition, and the mean intensity of the ROI was measured using ImageJ software. Statistics are listed in each figure legend.

## Acknowledgements

We thank John Dingus, Sophia Zhao, and Paula Montero-Llopis (Microscopy Resources on the North Quad) of Harvard Medical School, Xiaomei Sun and Peimin Ma of Lingang Laboratory, Li Tan and Kangning Sang (Optical Imaging Core Facility) of Shanghai Research Center for Brain Science and Brain-Inspired Intelligence for advisory and technical support. This work was funded by Howard Hughes Medical Institute (to CLC), National Institutes of Health (NIH) grants K99EY030951 (to YX before June 30, 2022), Lingang Laboratory startup fund (to YX after July 20, 2022).

## Additional information

### Funding

| Funder | Grant reference number | Author |
|---|---|---|
| National Eye Institute | K99EY030951 | Yunlu Xue |
| Howard Hughes Medical Institute | | Constance L Cepko |
| Lingang Laboratory | Start up fund | Yunlu Xue |

The funders had no role in study design, data collection and interpretation, or the decision to submit the work for publication.

### Author contributions

Yunlu Xue, Conceptualization, Data curation, Formal analysis, Funding acquisition, Investigation, Visualization, Methodology, Writing – original draft, Writing – review and editing; Yimin Zhou, Formal analysis, Validation, Investigation, Visualization; Constance L Cepko, Conceptualization, Resources, Supervision, Funding acquisition, Writing – original draft, Project administration, Writing – review and editing

### Author ORCIDs

Yunlu Xue  http://orcid.org/0000-0002-2088-9826
Constance L Cepko  https://orcid.org/0000-0002-9945-6387

### Ethics

This study was performed in strict accordance with the recommendations in the Guide for the Care and Use of Laboratory Animals of the National Institutes of Health. All of the animals were handled according to approved institutional animal care and use committee (IACUC) protocols of the Harvard Medical Area (#IS00001695-3) and Lingang Laboratory (#NZXSP-2022-4). The protocol was approved by the Harvard Medical Area Standing Committee on Animals (assurance number: D16-00270) or IACUC committee of Lingang Laboratory.

Reviewer #1 (Public Review): https://doi.org/10.7554/eLife.90749.4.sa1
Reviewer #2 (Public Review): https://doi.org/10.7554/eLife.90749.4.sa2
Author response https://doi.org/10.7554/eLife.90749.4.sa3

## Additional files

### Supplementary files
• MDAR checklist

### Data availability

All data generated during this study are included in the manuscript and supporting files; source data files have been provided for all figures.

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
