## [Editor Report · eLife assessment]

This **fundamental** study advances our understanding of the cell specific treatment of cone photoreceptor degeneration by Txnip. The evidence supporting the conclusions is **compelling** with rigorous genetic manipulation of Txnip mutations. The work will be of broad interest to vision researchers, cell biologists and biochemists.

---

## [Referee Report · Reviewer #1 (Public Review)]

Summary:

This is a follow-up study to the authors' previous eLife report about the roles of an alpha-arrestin called protein thioredoxin interacting protein (Txnip) in cone photoreceptors and in the retinal pigment epithelium. The findings are important because they provide new information about the mechanism of glucose and lactate transport to cone photoreceptors and because they may become the basis for therapies for retinal degenerative diseases.

Strengths:

Overall, the study is carefully done and, although the analysis is fairly comprehensive with many different versions of the protein analyzed, it is clearly enough described to follow. Figure 4 greatly facilitated my ability to follow, understand and interpret the study. The authors have appropriately addressed a few concerns about statistical significance and the relationship between their findings and previous studies of the possible roles of Txnip on GLUT1 expression and localization on the surfaces of RPE cells.

---

## [Referee Report · Reviewer #2 (Public Review)]

The hard work of the authors is much appreciated. With overexpression of a-arrestin Txnip in RPE, cones and the combined respectively, the authors show a potential gene agnostic treatment that can be applied to retinitis pigmentosa. Furthermore, since Txnip is related to multiple intracellular signaling pathway, this study is of value for research in the mechanism of secondary cone dystrophy as well.

Strengths

- The follow-up study builds on innovative ground by exploring the impact of TxnipC247S and its combination with HSP90AB1 knockdown on cone survival, offering novel therapeutic pathways.

- Testing of different Txnip deletion mutants provides a nuanced understanding of its functional domains, contributing valuable insights into the mechanism of action in RP treatment.

- The findings regarding GLUT1 clearance and the differential effects of Txnip mutants on cone and RPE cells lay the groundwork for targeted gene therapy in RP.

Comments on revised version:

The researchers answered our questions and included additional discussion in the manuscript.

---

## [Author Response]

The following is the authors’ response to the previous reviews.

**Public Reviews:**

**Reviewer #1 (Public Review):**
Summary:This is a follow-up study to the authors' previous eLife report about the roles of an alpha-arrestin called protein thioredoxin interacting protein (Txnip) in cone photoreceptors and in the retinal pigment epithelium. The findings are important because they provide new information about the mechanism of glucose and lactate transport to cone photoreceptors and because they may become the basis for therapies for retinal degenerative diseases.Strengths:Overall, the study is carefully done and, although the analysis is fairly comprehensive with many different versions of the protein analyzed, it is clearly enough described to follow. Figure 4 greatly facilitated my ability to follow, understand and interpret the study. The authors have appropriately addressed a few concerns about statistical significance and the relationship between their findings and previous studies of the possible roles of Txnip on GLUT1 expression and localization on the surfaces of RPE cells.

We are delighted that Reviewer #1 is satisfied with this revised version.

**Reviewer #2 (Public Review):**
The hard work of the authors is much appreciated. With overexpression of a-arrestin Txnip in RPE, cones and the combined respectively, the authors show a potential gene agnostic treatment that can be applied to retinitis pigmentosa. Furthermore, since Txnip is related to multiple intracellular signaling pathway, this study is of value for research in the mechanism of secondary cone dystrophy as well.There are a few areas in which the article may be improved through further analysis and application of the data, as well as some adjustments that should be made in to clarify specific points in the article.StrengthsThe follow-up study builds on innovative ground by exploring the impact of TxnipC247S and its combination with HSP90AB1 knockdown on cone survival, offering novel therapeutic pathways.Testing of different Txnip deletion mutants provides a nuanced understanding of its functional domains, contributing valuable insights into the mechanism of action in RP treatment.The findings regarding GLUT1 clearance and the differential effects of Txnip mutants on cone and RPE cells lay the groundwork for targeted gene therapy in RP.WeaknessesThe focus on specific mutants and overexpression systems might overlook broader implications of Txnip interactions and its variants in the wider context of retinal degeneration.

Txnip is not expressed in WT or RP cones, as described in our previous study (Xue et al., 2021, eLife), so we could not perform loss of function assays. We thus chose overexpression, and assayed various alleles, based upon the literature, as we describe in our manuscript.

The study's reliance on cell count and GLUT1 expression as primary outcomes misses an opportunity to include functional assessments of vision or retinal health, which would strengthen the clinical relevance.

In our previous study, we demonstrated that the optomotor response of Txnip-treated RP mice improved (Xue et al., 2021, eLife). Also, as described in our previous Txnip study, as well as an independent study (Xue et al., 2021, eLife; Xue et al., 2023, PNAS), ERG assays of Txnip-treated RP cones were no different than the controls. Other therapies that prolong RP cone survival and the optomotor response in our lab also failed to save the ERG, suggesting that there are other pathways that need to be addressed, e.g. the visual cycle. A combination therapy addressing multiple problems is one of our goals.

The paper could benefit from a deeper exploration of why certain treatments (like Best1-146 Txnip.C247S) do not lead to cone rescue and the potential for these approaches to exacerbate disease phenotypes through glucose shortages.

This system is more complicated than we currently understand, and more work needs to be done.

Minor inconsistencies, such as the missing space in text references and the need for clarification on data representation (retinas vs. mice), should be addressed for clarity and accuracy.

The missing spaces are added.

We described the strategy of injecting the same mouse in each eye, one eye with control and one with the experimental vector. However, the following sentence has been added to the Materials and Methods to better assist the reader:

“In almost all experiments, other than as noted, one eye of the mouse was treated with control (AAV8-RedO-H2BGFP, 2.5 × 108 vg/eye), and the other eye was treated with the experimental vector plus AAV8-RedO-H2BGFP, 2.5 × 108 vg/eye.”

The observation of promoter leakage and potential vector tropism issues raise questions about the specificity and efficiency of the gene delivery system, necessitating further discussion and validation.

The following sentences have been added to the Results. We do not think this phenomenon affects the practice of the experiments or the interpretation of the results in this study.

“To enable automated cone counting and trace the infection, we co-injected an AAV (AAV8-RedO-H2BGFP-WPRE-bGHpA) encoding an allele of GFP fused to histone 2B (H2BGFP), which localized to the nucleus. As the red opsin promoter was used to express this gene, H2BGFP was seen in cone nuclei, but not in the RPE, if AAV8-RedO-H2BGFP-WPRE-bGHpA was injected alone. However, when an AAV that expressed in the RPE, i.e. AAV8-Best1-Sv40intron-(Gene)-WPRE-bGHpA, was co-injected with AAV8-RedO-H2BGFP-WPRE-bGHpA, H2BGFP was expressed in the RPE, along with expression in cones (Figure 2A). We speculate that this is due to concatenation or recombination of the two genomes, such that the H2BGFP comes under the control of the RPE promoter. This may be due to the high copy number of AAV in the RPE, as it did not happen in the reverse combination, i.e. AAV with an RPE promoter driving GFP and a cone promoter driving another gene, perhaps due to the observation that the AAV genome copy number is »10 fold lower in cones than in the RPE (Wang et al., 2020).”